# Variational prompt tuning improves generalization of vision-language models

## Abstract

Prompt tuning provides an efficient mechanism to adapt large vision-language models to downstream tasks by treating part of the input language prompts as learnable parameters while freezing the rest of the model. Existing works for prompt tuning are however prone to damaging the generalization capabilities of the foundation models, because the learned prompts lack the capacity of covering certain concepts within the language model. To avoid such limitation, we propose a probabilistic modeling of the underlying distribution of prompts, allowing prompts within the support of an associated concept to be derived through stochastic sampling. This results in a more complete and richer transfer of the information captured by the language model, providing better generalization capabilities for downstream tasks. The resulting algorithm relies on a simple yet powerful variational framework that can be directly integrated with other developments. We show our approach is seamlessly integrated into both standard and conditional prompt learning frameworks, improving the performance on both cases considerably, especially with regards to preserving the generalization capability of the original model. Our method provides the current state-of-the-art for prompt learning, surpassing CoCoOp by 1.6% average Top-1 accuracy on the standard benchmark. Remarkably, it even surpasses the original CLIP model in terms of generalization to new classes. Implementation code will be released.

## 1 Introduction

In a continuous quest for better pre-training strategies, models based on image and language supervision have set impressive milestones, with CLIP (Radford et al., 2021), ALIGN (Jia et al., 2021) and Flamingo (Alayrac et al., 2022) being leading examples. Contrastively trained vision-language models consist of image and text encoders that align semantically-related concepts in a joint embedding space. Such models offer impressive zero-shot image classification by using the text encoder to generate classifier weights from arbitrarily newly defined category classes without relying on any visual data. In particular, the class name is used within a handcrafted *prompt* template and then tokenized and encoded into the shared embedding space to generate new classifier weights. Rather than manually defining prompts, Zhou et al. (2022b) and Lester et al. (2021) proposed that prompts can be instead optimized in a data-driven manner through back-propagation by maximizing a cross-entropy loss on the downstream task. However, despite the performance improvement on downstream tasks, prompt learning negatively affects the generalization capability of the vision-language model.

While subsequent works have focused on how to bridge the generalization gap, e.g. Zhou et al. (2022a); Zhu et al. (2022), in practice the generalization power of the foundation model is significantly degraded. Our work tackles this same problem, as it seeks to improve downstream performance but without degrading the generalization capability of the original model.

To do so, we propose a data-driven method for directly learning the underlying distribution within the prompt space associated to the target concept. In particular, we frame prompt tuning as a variational inference problem, where a base learned prompt is combined with a residual vector sampled from the instance-specific underlying distribution. This formulation provides two advantages. First, it investigates the prompt space more thoroughly and results in more informative use of the language space, leading to better generalization. Second, it enables us to boost performance by capturing the uncertainty information in fine-grained classification problems. The resulting approach is orthogonal

to standard prompt learning approaches, being effective when combined with both standard (Zhou et al., 2022b) and conditional (Zhou et al., 2022a) approaches. In fact, when combined with the conditional approach, our method maintains the gains on the seen classes provided by the conditional method while simultaneously matching or even surpassing the generalization capability on unseen classes of the original vision-language model.

In summary, our contributions in this paper are as follows:

1. We propose a variational framework that is capable of capturing the general or instance specific distribution within the prompt space. Since generalization is obtained through transfer from the language space, we obtain better generalization capability.

2. We show that the proposed approach is orthogonal to recent developments, and can be successfully combined with both standard and conditional prompt learning variants.

3. We empirically show that our proposed method improves performance and provides better generalization, leading to state-of-the-art accuracy in 24 out of 28 standard benchmarks set forth by prior work, surpassing CoCoOp by 1.6% average Top-1 accuracy.

## 2 RELATED WORKS

**Prompt learning in NLP.** Prompt learning was originally proposed within the NLP domain, following the appearance of foundation models such as GPT-3 (Brown et al., 2020). Early prompt learning methods constructed prompts by combining words in the language space such that the model would perform better on downstream evaluation (Shin et al., 2020; Jiang et al., 2020). Subsequent methods, e.g. Li & Liang (2021); Lester et al. (2021), prepend a set of learnable prompts to the input of a frozen model and optimize through back-propagation, which allows better flexibility than using existing words, at the cost of leading to prompts that do not correspond to an actual phrase. Instead, He et al. (2022) focus on a multi-task scenario and use a HyperNetwork to conditionally generate task-specific and layer-specific prompts that are pre-pended to the values and keys inside the self-attention layers of a the frozen model. Within the NLP domain, prompt learning has also been shown to work better than in-context learning (Liu et al., 2022).

**Prompting in Vision and Language models.** Research on prompt learning for vision-language models have been largely inspired by prior work within NLP. Similar to e.g. Li & Liang (2021), CoOp (Zhou et al., 2022b) proposes a prompt learning method that optimizes unified or class specific prompts in the continuous space through back-propagation. While CoOp obtains good accuracy on downstream tasks, it negatively affects the generalization ability to new unseen classes. Co-CoOp (Zhou et al., 2022a) extends CoOp and partially bridges the generalization gap by generating instance-specific prompt residuals through a conditioning mechanism dependent on the visual data. ProGrad (Zhu et al., 2022) shares the same goal as CoCoOp of bridging the generalization gap, but instead proposes to match the gradient of the prompt to the general knowledge of the CLIP model to prevent prompt tuning from forgetting the general knowledge learned from the foundation model. Alternative directions consist of test-time prompt tuning (Shu et al., 2022), where consistency across multiple views is used as the supervisory signal, and unsupervised prompt learning (Huang et al., 2022), where a pseudo-labelling strategy is proposed instead to obtain the labels needed to drive the prompt learning. Perhaps the most similar work to ours is Lu et al. (2022). In this work, the authors use an ensemble of prompts and model their distribution within the language embedding space, with optimization seeking to minimize the negative log-likelihood with respect to the corresponding visual embedding. Unlike ours, their method relies on hand-crafted rules to define the prompt ensemble, thus still relying on the effectiveness of hand-crafted designs. The number of learnable prompts is also pre-defined, potentially offering sub-optimal coverage of an NLP concept. Finally, it is not clear how to apply their strategy within the context of conditional prompt learning. We believe that modelling the input prompt space rather than relying on a fixed number of templates is a more powerful and flexible approach. We provide empirical evidence of the superiority of our approach in the experiments.

While beyond our current scope, it is worth noting that prompt learning has been applied to a wider range of problems and scenarios, which highlights its power and flexibility. Among them are important topics such as unsupervised domain adaptation (Ge et al., 2022), multi-label classification (Sun et al., 2022), video classification (Ju et al., 2022), object detection (Du et al., 2022; Feng et al., 2022)

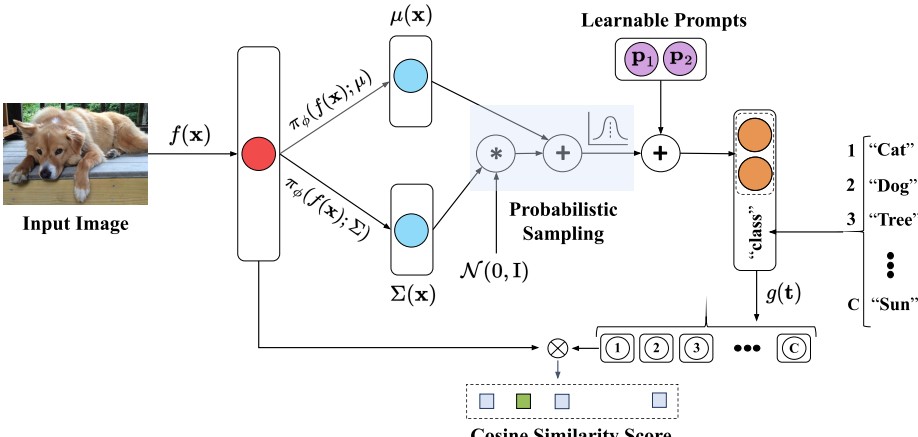

Figure 1: **Overview of variational prompt tuning**. For each input image $\mathbf{x}$, we use image features $f(\mathbf{x})$ to infer the mean $\mu(\mathbf{x})$ and standard deviation $\Sigma(\mathbf{x})$ of the residual distribution using metanet $\pi_\phi$. The prompts to generate the classifier weights are constructed by summing up learnable prompts $\mathbf{p}$ and residual samples from the residual distribution. The obtained prompts are fed through a text encoder $g(\mathbf{t})$, and the classifier weights are estimated. Finally, the cosine similarity scores are computed between the image features $f(\mathbf{x})$ and the classifier weights.

or pixel-level labelling (Rao et al., 2022). Finally, prompt learning as a means to adapt pre-trained models has also been applied to purely vision models (Jia et al., 2022; Sandler et al., 2022) providing similar performance to fine tuning the whole model but with great parameter efficiency.

## 3 METHOD

### 3.1 BACKGROUND

**Contrastive Language-Image Pretraining (CLIP)** (Radford et al., 2021) consists of an *image encoder* $f(\mathbf{x})$ and *text encoder* $g(\mathbf{t})$, each producing a $d$-dimensional ($L_2$ normalized) embedding from an arbitrary image $\mathbf{x} \in \mathbb{R}^{3 \times H \times W}$, and word embeddings $\mathbf{t} \in \mathbb{R}^{L \times e}$, with $L$ representing the text length and $e$ the embedding dimension[1]. Both encoders are trained together using a contrastive loss from a large-scale dataset composed of paired images and captions. Once trained, CLIP can be used for zero-shot $C$-class image classification by generating each of the $c$ classifier weights $\mathbf{w}_c$ as the $d$-dimensional text encoding $g(\mathbf{t}_c)$. Here $\mathbf{t}_c$ results from adding the class-specific word embedding $\mathbf{e}_c$ to a pre-defined prompt $\mathbf{p} \in \mathbb{R}^{L-1 \times e}$, i.e., $\mathbf{w}_c = g(\mathbf{t}_c)$ with $\mathbf{t}_c = \{\mathbf{p}, \mathbf{e}_c\}$. The prompt $\mathbf{p}$ is manually crafted to capture the semantic meaning of the downstream task, e.g., $\mathbf{t}_c =$ "An image of a {class}". The probability of image $\mathbf{x}$ being classified as $y \in \{1...C\}$ is thus defined as $p(y|\mathbf{x}) = \frac{e^{f(\mathbf{x})^T \mathbf{w}_y}}{\sum_c^C e^{f(\mathbf{x})^T \mathbf{w}_c}}$.

**Context Optimization (CoOp)** (Zhou et al., 2022b) provides a learned alternative to manually defining prompts. CoOp learns a fixed prompt from a few annotated samples. The prompt is designed as a learnable embedding matrix $\mathbf{p} \in \mathbb{R}^{L \times e}$ which is updated via back-propagating the classification error through the frozen CLIP model. Specifically, for a set of $N$ annotated meta-training samples $\{\mathbf{x}_i, y_i\}_{i=1}^N$, the prompt $\mathbf{p}$ is obtained by minimizing the cross-entropy loss, as:

$$\mathbf{p}^* = \arg\min_{\mathbf{p}} \mathbb{E}_{\mathbf{x}_i, y_i}[-\log p(y_i|\mathbf{x}_i, \mathbf{p})]. \tag{1}$$

Note that this approach, while resembling that of common meta-learning approaches, can still be deployed in a zero-shot scenario provided that for new classes the classification weights will be given by the text encoder. Although this approach generalizes to new tasks with few training iterations, learning a fixed prompt is sensitive to domain shifts between the annotated samples and the test set.

---

[1]In CLIP the word embedding is learned together with the text encoder. A *tokenizer* is used to convert the text into one-hot vectors, or tokens, that can be directly mapped into the word embeddings. For the sake of clarity we refer indistinctly to words and word embeddings.

**Conditional Prompt Learning (CoCoOp)** (Zhou et al., 2022a) attempts to overcome domain shifts by learning an instance-specific continuous prompt that is conditioned on the input image. To ease the training of a conditional prompt generator, CoCoOp defines each conditional token in a residual way, with a task-specific, learnable set of tokens $\mathbf{p}$ and a residual vector that is conditioned on the input image. Assuming $\mathbf{p}$ to be composed of $L$ learnable tokens $\mathbf{p}=[\mathbf{p}_1, \mathbf{p}_2, ..., \mathbf{p}_L]$, the residual vector $\mathbf{r}(\mathbf{x})=\pi_\phi(f(\mathbf{x})) \in \mathbb{R}^e$ is produced by a small neural network $\pi_\phi$ with as input the image features $f(\mathbf{x})$. The new prompt is then computed as $\mathbf{p}(\mathbf{x})=[\mathbf{p}_1 + \mathbf{r}(\mathbf{x}), \mathbf{p}_2 + \mathbf{r}(\mathbf{x}), ..., \mathbf{p}_L + \mathbf{r}(\mathbf{x})]$. The training now comprises learning the task-specific prompt $\mathbf{p}$ and the parameters $\phi$ of the neural network $\pi_\phi$. Defining the context-specific text embedding $\mathbf{t}_c(\mathbf{x})=\{\mathbf{p}(\mathbf{x}), \mathbf{e}_c\}$, and $p(y|\mathbf{x})$ as :

$$p(y|\mathbf{x}) = \frac{e^{f(\mathbf{x})^T g(\mathbf{t}_c(\mathbf{x}))}}{\sum_c^C e^{f(\mathbf{x})^T g(\mathbf{t}_c(\mathbf{x}))}}, \tag{2}$$

the learning is formulated as:

$$\mathbf{p}^*, \phi^* = \arg \min_{\mathbf{p},\phi} \mathbb{E}_{\mathbf{x}_i,y_i}[-\log p(y_i|\mathbf{x}_i, \mathbf{p}, \phi)]. \tag{3}$$

While CoCoOp achieves state-of-the-art results in a large variety of downstream tasks, it is still prone to the domain shift problem, considering that $\pi_\phi$ provides a deterministic residual vector from the image features $f(\mathbf{x})$ which are expected to be domain-specific.

**Prompt Distribution Learning (ProDA)** (Lu et al., 2022) is work concurrent to CoCoOp that focuses learning a distribution of prompts that generalize to a broader set of tasks. ProDA proposes to learn a collection of prompts $\mathbf{P}=\{\mathbf{p}_k\}_{k=1}^{\mathcal{K}}$ that can be used to subsequently generate an *a posteriori* distribution of the classifier weights for each of the target classes. For a given mini-batch of $K$ sampled prompts $\mathbf{p}_k \sim \mathbf{P}$, the classifier weights $\mathbf{w}_c$ are sampled from the posterior distribution $\mathcal{N}(\mu_{\mathbf{w}_{1:C}}, \Sigma_{\mathbf{w}_{1:C}})$, with mean $\mu_{\mathbf{w}_{1:C}}$ and covariance $\Sigma_{\mathbf{w}_{1:C}}$ computed from the collection $\{\mathbf{w}_{k,c} = g(\mathbf{t}_{k,c})\}_{c=1:C,k=1:K}$, with $\mathbf{t}_{k,c} = \{\mathbf{p}_k, \mathbf{e}_c\}$. The objective is now formulated as:

$$\mathbf{P}^* = \arg \min_{\mathbf{P}} \mathbb{E}_{\mathbf{x}_i,y_i}[-\log \mathbb{E}_{\mathbf{w}_l \sim \mathcal{N}(\mu_{\mathbf{w}_{1:C}}, \Sigma_{\mathbf{w}_{1:C}})} p(y_i|\mathbf{x}_i, \mathbf{w}_l)]. \tag{4}$$

Computing $\mathbb{E}_{\mathbf{w}_l} p(y_i|\mathbf{x}_i, \mathbf{w}_l)]$ is intractable and an upper bound to Eq. 4 is derived. During inference, the classifier weights are set to those given by the predictive mean $\mathbf{w}_c = \mu_{\mathbf{w}_{1:C}}$, computed across the collection of learned prompts $\mathbf{P}$. While showing promising results compared to CoOp, how to combine it with the conditional prompt learning framework of CoCoOp is unclear.

## 3.2 Variational Prompt Tuning

In this paper, we propose to model the input prompt space in a probabilistic manner, as an *a priori, instance-specific* distribution. In particular, we define a distribution $p_\gamma$ over the prompts $\mathbf{p}$ that is instance-specific, i.e. $\mathbf{p} \sim p_\gamma(\mathbf{x})$. To this end, we assume that $\mathbf{p}$ can be split into a fixed set of prompts $\mathbf{p}_i$ and an instance specific residual vector $\mathbf{r}$ that act as a latent variable over $\mathbf{p}$. The instance-specific prompt is then defined as:

$$\mathbf{p}_\gamma(\mathbf{x}) = [\mathbf{p}_1 + \mathbf{r}_\gamma, \mathbf{p}_2 + \mathbf{r}_\gamma, \cdots, \mathbf{p}_L + \mathbf{r}_\gamma], \ \mathbf{r}_\gamma \sim p_\gamma(\mathbf{x}), \tag{5}$$

where $p_\gamma(\mathbf{x})$ refers to the real posterior distribution over $\mathbf{r}$ conditioned on the observed features $\mathbf{x}$. Denoting the class-specific input as $\mathbf{t}_{c,\gamma}(\mathbf{x})=\{\mathbf{p}_\gamma(\mathbf{x}), \mathbf{e}_c\}$, the marginal likelihood $p(y|\mathbf{x})$ is:

$$p(y|\mathbf{x}) = \int_\gamma \frac{e^{f(\mathbf{x})^T g(\mathbf{t}_{c,\gamma}(\mathbf{x}))}}{\sum_{c'} e^{f(\mathbf{x})^T g(\mathbf{t}_{c',\gamma}(\mathbf{x}))}} p(\mathbf{p}_\gamma(\mathbf{x})) d\gamma. \tag{6}$$

Solving Eq. 3 with the marginal likelihood defined as in Eq. 6 is intractable, as it requires computing $p_\gamma(\mathbf{r}|\mathbf{x})p_\gamma(\mathbf{x})$. Instead, we resort to deriving a lower bound, by introducing a variational posterior distribution $\pi_\phi(\mathbf{x})$ from which the residual $\mathbf{r}_\gamma$ can be sampled. The variational bound is defined as:

$$\log p(y|\mathbf{x}) \geq \mathbb{E}_{\pi_\phi(\mathbf{r}|\mathbf{x})}[\log p(y|\mathbf{x}, \mathbf{r})] - D_{\mathrm{KL}}[\pi_\phi(\mathbf{r}|\mathbf{z})\|p_\gamma(\mathbf{r})], \tag{7}$$

with $p(y|\mathbf{x}, \mathbf{r}) \propto e^{f(\mathbf{x})^T g(\mathbf{t}_{c,\gamma}(\mathbf{x}))}$, where the dependency on $\mathbf{r}$ comes through the definition of $\mathbf{t}_{c,\gamma}$. The variational posterior distribution $\pi_\phi$ plays a role akin to the metanet in CoCoOp. We thus refer to it as metanet in the following to align terminology. Following standard variational optimization practices (Kingma & Welling, 2014; Gordon et al., 2019), we define $\pi_\phi$ as a Gaussian distribution

conditioned on the input image features $\mathbf{x}$, as $\mathbf{r}(\mathbf{x}) \sim \mathcal{N}(\mu(\mathbf{x}), \Sigma(\mathbf{x}))$, with $\mu$ and $\Sigma$ parameterized by two linear layers placed on top of the metanet $\pi_\phi$ (see Figure 1). The prior $p_\gamma(\mathbf{r})$ is defined as $\mathcal{N}(\mathbf{0}, \mathbf{I})$, and we make use of the reparameterization trick to generate Monte-Carlo samples from $\pi_\phi$ to maximize the right side of Eq. 7. The optimization of Eq. 7 comprises learning the prompt embeddings $\{\mathbf{p}_i\}_{i=1}^L$ as well as the parameters of the metanet $\pi_\phi$ and the linear layers parameterizing $\mu$ and $\Sigma$. Note that this adds little complexity as it requires learning $\mathbf{p}$ and $\pi_\phi$, given that $\mu$ and $\Sigma$ are defined as two linear layers on top of $\pi_\phi$.

**Inference.** At test time, $K$ residuals are sampled from the conditional distribution $\pi_\phi(\mathbf{x})$, which are used to generate $K$ different prompts per class $\mathbf{p}_k = [\mathbf{p}_1 + \mathbf{r}_k, \mathbf{p}_2 + \mathbf{r}_k, \cdots, \mathbf{p}_L + \mathbf{r}_k]$. Each prompt is prepended to the class-specific embedding to generate a series of $K$ separate classifier weights $\mathbf{w}_{k,c}$. We then compute $p(y=c\,|\,\mathbf{x}) = (1/K) \sum_{k=1}^K p(y=c\,|\,\mathbf{x}, \mathbf{w}_{k,c})$ and select $\hat{c} = \arg\max_c p(y=c\,|\,\mathbf{x})$ as the predicted class. It is worth noting that because the posterior distribution is generated by the text encoder, it is not expected that for $K \rightarrow \infty$, $(1/K) \sum_k p(y=c\,|\,\mathbf{x}, \mathbf{w}_{c,k}) \rightarrow p(y=c|\mathbf{x}, g(\{\mu(\mathbf{x}), \mathbf{e}_c\}))$, meaning that sampling at inference time remains relevant. We study the dependency on the number of samples in the ablations.

Notably, our framework can be also used with the unconditional setting of CoOp, by simply removing the dependency of the input image from the latent distribution. In such scenario, we keep a fixed set of prompt embeddings and learn a global latent distribution $p_\gamma$ over the residual vectors $\mathbf{r}$, as $\mathbf{r} \sim \mathcal{N}(\mu, \Sigma)$, where $\mu$ and $\Sigma$ are parameterized by two learnable vectors. In this case, $p_\gamma$ is a general distribution learned during training with no dependency on the input sample $\mathbf{x}$. We show that CoOp with our proposed approach improves generalization on new classes.

## 4 EXPERIMENTS AND RESULTS

### 4.1 EXPERIMENTAL SETUP

We follow the exact experimental setup of CoCoOp (Zhou et al., 2022a), currently the state-of-the-art for prompt tuning for vision-language models. We describe the setup in the following.

**Three tasks and fifteen datasets.** We evaluate variational prompt tuning for three different tasks: *base-to-new generalization*, *cross-dataset transfer*, and *cross-domain generalization*. For base-to-new generalization and cross-dataset transfer tasks, we rely on the same 11 image recognition datasets as Zhou et al. (2022b;a). These include generic image classification datasets (ImageNet by (Deng et al., 2009) and Caltech101 by (Fei-Fei et al., 2004)), fine-grained classification datasets (OxfordPets by (Parkhi et al., 2012), StanfordCars by (Krause et al., 2013), Flowers102 by (Nilsback & Zisserman, 2008), Food101 by (Bossard et al., 2014) and FGVCAircraft by (Maji et al., 2013)), scene recognition (SUN397 by (Xiao et al., 2010)), action recognition (UCF101 by (Soomro et al., 2012)), texture classification (DTD by (Cimpoi et al., 2014)), and satellite imagery recognition (EuroSAT by (Helber et al., 2019)). For the cross-domain generalization task, we train our model on ImageNet and report on ImageNetV2 (Recht et al., 2019), ImageNet-Sketch (Wang et al., 2019), ImageNet-A (Hendrycks et al., 2021b), and ImageNet-R (Hendrycks et al., 2021a).

**Evaluation metrics.** We report average accuracy and harmonic mean $H = 2 \times (\text{base} \times \text{new})/(\text{base} + \text{new})$ (Xian et al., 2017) for the base-to-new generalization tasks. For cross-dataset transfer learning and domain adaptation, we provide average accuracy results.

**Baselines.** We compare against zero-shot CLIP (Radford et al., 2021), CoOp (Zhou et al., 2022b), CoCoOp (Zhou et al., 2022a), and ProDA (Lu et al., 2022). For zero-shot CLIP, CoOp, CoCoOp, all results are adopted from (Zhou et al., 2022a), and we reproduce all results for ProDA.

**Implementation details.** Our variational prompt tuning contains three sub-networks: an image encoder $f(\mathbf{x})$, a text encoder $g(\mathbf{t})$, and a metanet $\pi_\phi$. The image encoder $f(\mathbf{x})$ and text encoder $g(\mathbf{t})$ are a ViT-B/16 (Dosovitskiy et al., 2021) and transformer (Vaswani et al., 2017), which are initialized with CLIP's pre-trained weights and kept frozen during training, as in Zhou et al. (2022b;a). The metanet $\pi_\phi$ consists of two linear layers followed by ELU activation function as trunk and two linear heads on top to estimate the $\mu$ and $\Sigma$ of the residual distribution. For each task and dataset, we optimize the number of samples $K$ and epochs. Other hyper-parameters as well as the training pipeline in terms of few-shot task definitions are identical to Zhou et al. (2022b;a) (see table 6 and 7 in the appendix). Implementation code will be released.

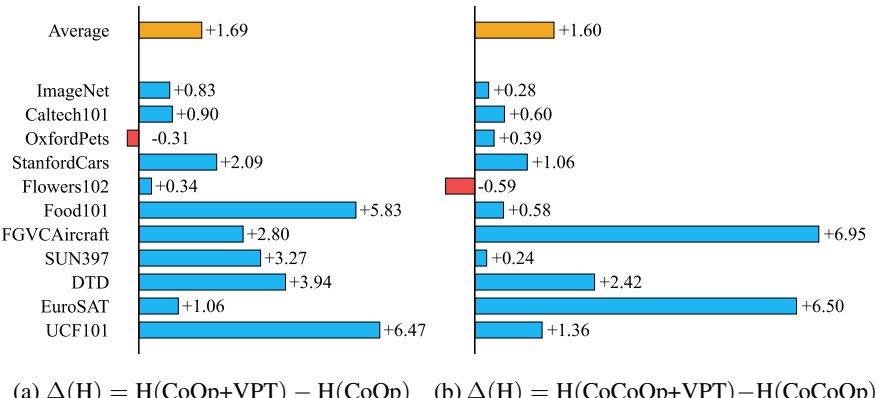

(a) $\Delta(\text{H}) = \text{H}(\text{CoOp+VPT}) - \text{H}(\text{CoOp})$    (b) $\Delta(\text{H}) = \text{H}(\text{CoCoOp+VPT}) - \text{H}(\text{CoCoOp})$

Figure 2: **Relative enhancement of variational prompt tuning** over CoOp and CoCoOp in terms of harmonic mean over 11 datasets for 3 distinct random seeds. Variational prompt tuning improves the harmonic mean for all baselines other than OxfordPets for CoOp and Flowers102 for CoCoOp.

## 4.2 BASE-TO-NEW GENERALIZATION

**Setup.** We report the few-shot generalization of our method on 11 datasets for three different random seeds. Each dataset is divided into two disjoint subsets: base classes and new classes. We train our method on base classes and evaluate it on both base and new classes. For a fair comparison, we follow Zhou et al. (2022a;a) in terms of dataset split and number of shots (see Table 6 appendix.)

**Results.** From the results in Table 1, the lack of generalization capability of the CoOp approach is evidenced by the considerable discrepancy between the base and new classes' accuracy. This is expected as CoOp only observes a small number of training samples to adapt the CLIP model for downstream tasks, resulting in overfitting. Adding variational prompt tuning to CoOp reduces overfitting to base classes, improving the performance of new classes by $11.54\%$ and the harmonic mean by $1.7\%$ across all datasets, at the expense of a base classes accuracy drop by $10.7\%$. Figure 2(a) depicts the relative improvement of our proposed strategy when compared to CoOp in terms of the harmonic mean, where we observe an improvement in 10 out of 11 datasets. The limited generalization capability of CoOp is mitigated by CoCoOp by exploiting instance-conditional prompts, which improves the accuracy on new classes from $63.22\%$ to $72.23\%$. Nonetheless, augmenting CoCoOp with variational prompts still improves performance on the new classes and the harmonic mean by $3.25\%$ and $1.6\%$, respectively, with a small decrease of $0.37\%$ in base accuracy. Note here however that training for longer increases the performance on base classes and lowers it on new classes. Our setting optimizes harmonic mean, but a simple tweak to the training scheduler on EuroSAT is enough to surpass CoCoOp on base class accuracy, see Table 8 on appendix A.2.2. Figure 2(b) shows per-dataset relative harmonic mean improvement. We observe an improvement in 10 out of 11 datasets. Note that the drop in the average base accuracy for CoCoOp+VPT is negligible compared with CoOp+VPT . Moreover, our best performing model CoCoOp+VPT performs better than ProDA (Lu et al., 2022) in terms of new classes accuracy and harmonic mean by $\%2.64$ and $\%0.78$. We also considered ensemble learning the soft prompt from different ways of initialization, this does not lead to an improvement (see Appendix A.2.3). It is also worth noting that CoOp+VPT and CoCoOp+VPT excel at CLIP zero-shot performance in base, new and harmonic accuracies.

## 4.3 CROSS-DATASET TRANSFER LEARNING

**Setup.** For cross-dataset transfer learning, the model is trained on a source dataset (ImageNet) and then assessed on 10 distinct target datasets. This experiment tries to determine how effectively our methods generalizes transfer beyond the scope of a single dataset.

**Results.** As reported in Table 2, CoOp+VPT has a drop in performance in ImageNet by $1.76\%$ while outperforming the target dataset on average by $1.63\%$. Moreover, our proposed method leads to an increase on 9 out of 10 target datasets, with a small drop in accuracy of $0.03\%$ in Caltech101. Note that on target datasets such as FGVCAircraft and DTD, our proposed method achieves an improvement of more than $3\%$. Similarly to CoOp, augmenting CoCoOp with our method still leads to an

Table 1: **Base-to-new generalization** comparison between the state-of-the-art and variational prompt tuning. We average our accuracy over three random seeds. Our proposed model is trained on a few-shot training set (base) and then evaluated on held-out classes (new). As shown, CoOp and CoCoOp overfit on base classes and do not provide good generalization on new classes. However, our model provides better generalization performance on new classes as well as harmonic mean.

| Dataset | | CLIP | CoOp | +VPT | Δ | CoCoOp | +VPT | Δ | ProDA |
|---|---|---|---|---|---|---|---|---|---|
| Average | Base | 69.34 | 82.66 | 71.98 | −10.71 | 80.47 | 80.10 | −00.37 | 81.56 |
| | New | 74.22 | 63.22 | 74.76 | +11.54 | 71.69 | 74.94 | +03.25 | 72.30 |
| | H | 71.69 | 71.65 | 73.34 | **+01.69** | 75.83 | 77.43 | **+01.60** | 76.65 |
| ImageNet | Base | 72.43 | 76.14 | 74.73 | −01.41 | 75.98 | 76.00 | +00.02 | 75.40 |
| | New | 68.14 | 67.88 | 70.60 | +02.72 | 70.43 | 70.93 | +00.50 | 70.23 |
| | H | 70.21 | 71.77 | 72.60 | **+00.83** | 73.10 | 73.37 | **+00.27** | 72.72 |
| Caltech101 | Base | 96.84 | 98.00 | 95.47 | −02.53 | 97.96 | 98.00 | +00.04 | 98.27 |
| | New | 94.00 | 89.81 | 93.80 | +03.99 | 93.81 | 94.93 | +01.12 | 93.23 |
| | H | 95.39 | 93.72 | 94.62 | **+00.90** | 95.84 | 96.44 | **+00.60** | 95.68 |
| OxfordPets | Base | 91.17 | 93.67 | 90.77 | −02.90 | 95.20 | 95.67 | +00.47 | 95.43 |
| | New | 97.26 | 95.29 | 97.83 | +02.54 | 97.69 | 98.00 | +00.31 | 97.83 |
| | H | 94.11 | 94.47 | 94.16 | −00.31 | 96.43 | 96.82 | **+00.39** | 96.62 |
| StanfordCars | Base | 63.37 | 78.12 | 65.27 | −12.85 | 70.49 | 72.93 | +02.44 | 74.70 |
| | New | 74.89 | 60.40 | 75.97 | +15.57 | 73.59 | 73.23 | −00.36 | 71.20 |
| | H | 68.65 | 68.12 | 70.21 | **+02.09** | 72.01 | 73.07 | **+01.06** | 72.91 |
| Flowers102 | Base | 72.08 | 97.60 | 72.97 | −24.63 | 94.87 | 95.70 | +00.83 | 97.70 |
| | New | 77.80 | 59.67 | 75.90 | +16.23 | 71.75 | 70.40 | −01.35 | 68.68 |
| | H | 74.83 | 74.06 | 74.40 | **+00.34** | 81.71 | 81.12 | **−00.59** | 80.66 |
| Food101 | Base | 90.10 | 88.33 | 90.37 | +02.04 | 90.70 | 91.03 | +00.33 | 90.30 |
| | New | 91.22 | 82.26 | 91.67 | +09.41 | 91.29 | 92.13 | +00.84 | 88.57 |
| | H | 90.65 | 85.18 | 91.01 | **+05.83** | 90.99 | 91.57 | **+00.58** | 89.43 |
| FGVCAircraft | Base | 27.19 | 40.44 | 29.57 | −10.87 | 33.41 | 34.40 | −00.99 | 36.90 |
| | New | 36.29 | 22.30 | 33.80 | +11.50 | 23.71 | 35.00 | +11.29 | 34.13 |
| | H | 31.08 | 28.74 | 31.54 | **+02.80** | 27.74 | 34.69 | **+06.95** | 35.46 |
| SUN397 | Base | 69.36 | 80.60 | 73.77 | −06.83 | 79.74 | 79.17 | −00.57 | 78.67 |
| | New | 75.35 | 65.89 | 77.90 | +12.01 | 76.86 | 77.87 | +01.01 | 76.93 |
| | H | 72.23 | 72.50 | 75.77 | **+03.27** | 78.27 | 78.51 | **+00.24** | 77.79 |
| DTD | Base | 53.24 | 79.44 | 57.67 | −21.77 | 77.01 | 75.30 | −01.71 | 80.67 |
| | New | 59.90 | 41.18 | 58.70 | +17.52 | 56.00 | 60.80 | +04.80 | 56.48 |
| | H | 56.37 | 54.24 | 58.18 | **+03.94** | 64.85 | 67.27 | **+02.42** | 66.44 |
| EuroSAT | Base | 56.48 | 92.19 | 67.97 | −24.22 | 87.49 | 80.30 | −07.19 | 83.90 |
| | New | 64.05 | 54.74 | 71.63 | +16.89 | 60.04 | 75.30 | +15.26 | 66.00 |
| | H | 60.02 | 68.69 | 69.75 | **+01.06** | 71.21 | 77.71 | **+06.50** | 73.88 |
| UCF101 | Base | 70.53 | 84.69 | 73.23 | −11.46 | 82.33 | 82.53 | +00.20 | 85.23 |
| | New | 77.50 | 56.05 | 74.63 | +18.58 | 73.45 | 75.77 | +02.32 | 71.97 |
| | H | 73.85 | 67.45 | 73.92 | **+06.47** | 77.64 | 79.00 | **+01.36** | 78.04 |

overall performance enhancement of 0.16% on 7 out of 10 target datasets, showing its effectiveness for cross-dataset transfer learning. In addition, unlike CoCoOp , which has better performance in ImageNet-like datasets such as Caltech101 and OxfordPets, our proposed method exhibits improvement on dissimilar datasets (e.g. FGVCAircraft, DTD, and EuroSAT), demonstrating its capacity to capture the unique characteristics of each dataset.

## 4.4 CROSS-DOMAIN GENERALIZATION

**Setup.** Lastly, We examine variational prompt tuning through the lens of distribution shift and robustness. We train our proposed model on the source dataset (ImageNet) for three different random seeds, and assess it on ImageNetV2, ImageNet-Sketch, ImageNet-A, and ImageNet-R. Prior work such as CoOp (Zhou et al., 2022b) and CoCoOp (Zhou et al., 2022a) demonstrate empirically that learning a soft-prompt improves the model's resilience against distribution shift and adversarial

Table 2: **Cross-dataset transfer learning** comparison between the state-of-the-art and our variational prompt tuning in terms of average accuracy on three different random seeds. Following (Zhou et al., 2022a), our proposed model is trained on a source dataset and evaluated on target datasets. As shown, variational prompt tuning performs better than other baselines 16 out of 20 datasets, although it loses performance on the source dataset.

| | Source | Target | | | | | | | | | | |
| Methods | ImageNet | Caltech101 | OxfordPets | StanfordCars | Flowers102 | Food101 | FGVCAircraft | SUN397 | DTD | EuroSAT | UCF101 | Average |
|---|---|---|---|---|---|---|---|---|---|---|---|---|
| CoOp | 71.51 | 93.70 | 89.14 | 64.51 | 68.71 | 85.30 | 18.47 | 64.15 | 41.92 | 46.39 | 66.55 | 63.88 |
| +VPT | 69.73 | 93.67 | 89.27 | 65.50 | 70.20 | 86.27 | 22.13 | 66.57 | 46.93 | 47.43 | 67.2 | 65.51 |
| Δ | −1.78 | −0.03 | +0.13 | +0.99 | +1.49 | +0.97 | +3.66 | +2.42 | +5.01 | +1.04 | +0.65 | +**1.63** |
| CoCoOp | 71.02 | 94.43 | 90.14 | 65.32 | 71.88 | 86.06 | 22.94 | 67.36 | 45.73 | 45.37 | 68.21 | 65.74 |
| +VPT | 70.70 | 93.67 | 90.63 | 65.00 | 70.90 | 86.30 | 24.93 | 67.47 | 46.10 | 45.87 | 68.67 | 65.95 |
| Δ | −0.32 | −0.76 | +0.49 | −0.32 | −0.98 | +0.24 | +1.99 | +0.11 | +0.37 | +0.50 | +0.46 | +**0.16** |

attack. Following their experiments, we are also interested in determining if treating prompts in a variational manner maintain or improve the performance.

**Results.** As reported in Table 3 our method enhances the accuracy of CoOp on ImageNet-Sketch, ImageNet-A, and ImageNet-R by $0.88\%$, $0.70\%$, and $2.19\%$ while degrading the performance on ImageNet and ImageNetV2 by $1.78\%$, by $1.03\%$. However, on CoCoOp, adding VPT, while losing the performance on source dataset similar to CoOp, consistently improves the accuracy on all target datasets which highlight the effectiveness of our proposed method.

Table 3: **Cross-domain Generalization** comparison between the state-of-the-art and variational prompt tuning in terms of average accuracy on three different random seeds. Our proposed model is trained on a source dataset and evaluated on target classes. Variational prompt tuning outperforms alternative baselines on the target datasets while losing performance on the source dataset.

| | | Source | Target | | | |
| Methods | Learnable | ImageNet | ImageNetV2 | ImageNet-Sketch | ImageNet-A | ImageNet-R |
|---|---|---|---|---|---|---|
| CLIP | ✗ | 66.73 | 60.83 | 46.15 | 47.77 | 73.96 |
| CoOp | ✓ | 71.51 | 64.20 | 47.99 | 49.71 | 75.21 |
| +VPT | ✓ | 69.73 | 63.17 | 48.87 | 50.77 | 77.40 |
| Δ | - | −1.78 | −1.03 | +0.88 | +0.70 | +2.19 |
| CoCoOp | ✓ | 71.02 | 64.07 | 48.75 | 50.63 | 76.18 |
| +VPT | ✓ | 70.70 | 64.23 | 49.20 | 51.33 | 77.00 |
| Δ | - | −0.32 | +0.16 | +0.45 | +0.70 | +0.82 |

## 4.5 ABLATIONS

**Effectiveness of the posterior distribution $q_\phi$.** We first ablate the effectiveness of the variational posterior distribution. To do this, we consider sampling one residual vector from the uniform distribution $\mathcal{U}(0,1)$, normal distribution $\mathcal{N}(0,\mathrm{I})$, normal distribution $\mathcal{N}(\mu(\mathbf{x}),0)$, normal distribution $\mathcal{N}(\mu(\mathbf{x}),\Sigma(\mathbf{x}))$, and report the new class accuracy for CoCoOp+VPT for one random seed in Table 4. Except for the EuroSAT dataset, a sample from the normal distribution $\mathcal{N}(\mu(\mathbf{x}),0)$ obtains the best-performingce in comparison with alternatives, showing that the mean of the normal distribution $\mu(\mathbf{x})$ is the most effective sample. In addition, we find that drawing one sample from $\mathcal{N}(\mu(\mathbf{x}),\Sigma(\mathbf{x}))$ yields superior results compared to drawing one sample from uniform distribution $\mathcal{U}(0,1)$ and normal distribution $\mathcal{N}(0,\mathrm{I})$, further demonstrating the efficacy of our proposed method in capturing the underlying distribution of the prompt space. We also ablate increasing the number of samples from the normal distribution $\mathcal{N}(\mu(\mathbf{x}),\Sigma(\mathbf{x}))$ to understand the informativeness of the learned variational distribution. It is shown that enlarging the number of samples further improves the model performance as they capture the prompt space appropriately.

**Prompt initialization.** We investigate the effectiveness of the prompt initialization on the new class accuracy for CoCoOp+VPT for one random seed. We consider two variants. In the first one, we ini-

Table 4: **Effectiveness of the posterior distribution**. The informative posterior distribution $\mathcal{N}(\mu(\mathbf{x}), \Sigma(\mathbf{x}))$ outperforms the two uninformative distributions $\mathcal{U}(0, 1)$ and $\mathcal{N}(0, \mathrm{I})$ by a large margin for all datasets. Increasing the number of samples further improves results.

| Methods | Samples | DTD | Flowers102 | EuroSAT | FGVCAircraft | UCF101 |
|---|---|---|---|---|---|---|
| $\mathcal{U}(0, 1)$ | 1 | 33.20 | 45.30 | 54.20 | 10.50 | 55.70 |
| $\mathcal{N}(0, \mathrm{I})$ | 1 | 26.60 | 36.10 | 50.00 | 07.70 | 48.80 |
| $\mathcal{N}(\mu(\mathbf{x}), 0)$ | 1 | 59.80 | 73.50 | 59.90 | 34.10 | 76.50 |
| $\mathcal{N}(\mu(\mathbf{x}), \Sigma(\mathbf{x}))$ | 1 | 56.40 | 72.30 | 64.50 | 33.00 | 75.60 |
| $\mathcal{N}(\mu(\mathbf{x}), \Sigma(\mathbf{x}))$ | 2 | 60.00 | 73.90 | 67.40 | 33.90 | 76.20 |
| $\mathcal{N}(\mu(\mathbf{x}), \Sigma(\mathbf{x}))$ | 5 | 62.20 | 74.00 | 71.00 | 34.20 | 76.60 |
| $\mathcal{N}(\mu(\mathbf{x}), \Sigma(\mathbf{x}))$ | 10 | 61.60 | 73.50 | 73.60 | 34.40 | 77.00 |

tialize the context tokens randomly using a normal distribution, whereas in second one we initialize the context tokens with "An image of a {class}". Table 5 summarizes this ablation. Comparing the two variants demonstrates that an appropriately initialized prompt consistently outperforms a randomly initialized prompt, highlighting the necessity for further research of the prompt space. We will leave it open for future research direction.

Table 5: **Prompt initialization.** Initializing the context tokens with an appropriate prompt "An image of a {class}" improves the performance compared to random tokens.

| Method | DTD | Flowers102 | EuroSAT | FGVCAircraft | UCF101 |
|---|---|---|---|---|---|
| Random | 74.10 | 67.23 | 75.10 | 34.30 | 81.30 |
| "An image of a {class}" | 74.80 | 70.05 | 77.90 | 35.50 | 82.50 |

**Number of Monte-Carlo samples.** When approximating the log-likelihood of input data, the number of Monte Carlo samples is an important hyperparameter. Generally, a large number of samples should lead to a better approximation and better classification accuracy. We ablate this hyperparameter on new accuracy for CoCoOp+VPT by varying the number of Monte Carlo samples at inference time. We show results for a varying number of samples in Figure 3 for DTD, Flowers102, EuroSAT, FGVCAircraft, and UCF101. Increasing the Monte Carlo samples from 1 to 10 consistently improves the new accuracy, afterwards the accuracy saturates. Hence, we recommend evaluating variational prompt tuning on a larger number of Monte Carlo samples for better model accuracy. To further aid the interpretation and understanding of the learned prompts we provide a further ablation on the variational distribution of our best performing model, CoCoOp+VPT in Appendix A.2.4 and A.2.5

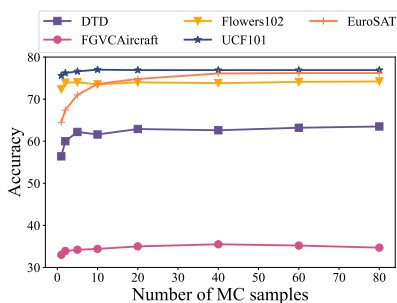

Figure 3: **Effectiveness of Monte Carlo samples** on new accuracy. As demonstrated, increasing the number of Monte Carlo samples boosts performance initially but reaches a plateau after 10 samples for all datasets.

## 5 CONCLUSION

In this paper, we introduce variational prompt tuning, a probabilistic modeling of the underlying distribution of prompts, allowing prompts within the support of an associated concept to be derived through stochastic sampling. By doing so, we are able to generate adaptive context tokens for each data point. This formulation leads to better generalization capabilities in terms of the new accuracy and harmonic mean for downstream tasks. We show that it can be seamlessly integrated into both standard and conditional prompt learning frameworks, considerably improving the performance in both cases. We conduct extensive experiments on 15 datasets and demonstrate the benefits of a variational formulation in learning data-driven prompts. Our method provides the current state-of-the-art for prompt learning, and constitutes, to the best of our knowledge, the first method for CLIP adaptation that fully maintains the generalization capability to new classes of the original model. Finally, prompt distribution can be obtained using other ways of density estimation, e.g. normalizing flows, energy-based models or diffusion models, and we will leave them open for future research.

## ETHICS STATEMENT

Our method has the potential to affect applications that often need rapid adaptation, such as medical imaging, astronomical imaging, and autonomous driving. Because of this, there may be negative societal consequences associated with the adoption of our technology. For example, a lack of fairness with models trained on insufficient data, regulatory compliance, patient privacy in medical imaging, and more general biases encoded into large vision-language models such as CLIP.

## REPRODUCIBILITY STATEMENT

Details on benchmarks, metrics, and the implementation of variational prompt tuning in terms of architecture and training are contained in Section 4. We also list of all hyperparameters in Table 6 and 7. Lastly, we will release the source code, scripts to reproduce the results and evaluating the performance of the model at: `https://github.com/<redacted>/<redacted>`.

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

# A APPENDIX

## A.1 HYPERPARAMETERS

In this section, we provide the detailed hyperparameter settings in Tables 6 and 7 that are used to generate results in the main paper for each dataset. There are two sets of hyperameters. The first are shared among the two variants of variational prompt tuning CoOp+VPT and CoCoOp+VPT (See Table 6). The second correspond to dataset-specific parameters that are optimized per dataset (See Table 7).

Table 6: Shared hyperparameters used to generate all results in the main paper.

| Hyperparameters | Values |
|---|---|
| Batch Size | 1 |
| Input Size | $224 \times 224$ |
| Input Interpolation Method | "Bicubic" |
| Input Mean | $[0.48145466, 0.4578275, 0.40821073]$ |
| Input STD | $[0.26862954, 0.26130258, 0.27577711]$ |
| Transformation | ["random resized crop", "random flip", "normalize"] |
| Optimizer | SGD |
| Learning Rate | $2e-3$ |
| LR Scheduler | "cosine" |
| Warmup Epoch | 1 |
| Warmup Type | "Constant" |
| Warmup LR | $1e-5$ |
| Backbone | ViT-B/16 |
| Prompt Length | 4 |
| Prompt Initialization | "a photo of a {class}" |
| Number of Shots | 16 |

Table 7: Dataset-specific hyperparameters used to generate all results in the main paper.

| Hyperparameters | ImageNet | Caltech101 | OxfordPets | StanfordCars | Flowers102 | Food101 | FGVCAircraft | SUN397 | DTD | EuroSAT | UCF101 |
|---|---|---|---|---|---|---|---|---|---|---|---|
| Number of Monte Carlo Samples | 10 | 20 | 40 | 20 | 10 | 20 | 10 | 20 | 40 | 20 | 5 |
| Number of Epochs | 10 | 20 | 20 | 40 | 40 | 20 | 10 | 10 | 10 | 60 | 20 |

## A.2 MORE ABLATIONS

### A.2.1 VISION ENCODER ALTERNATIVES.

All previous experiments benefit from ViT-B/16 as the vision encoder's backbone following (Zhou et al., 2022b;a; Lu et al., 2022). For completeness, in Figures 4, 5, and 6, we replace this vision encoder with a Resnet50 and Resnet100 and examine its impact on base accuracy, new accuracy and harmonic mean for one random seed. The visual transformer outperforms the Resnet alternatives across all 10 datasets for new accuracy and harmonic mean and in 9 out of 10 datasets for base accuracy. Hence, we suggest training and evaluating variational prompt tuning on visual transformer for better model performance.

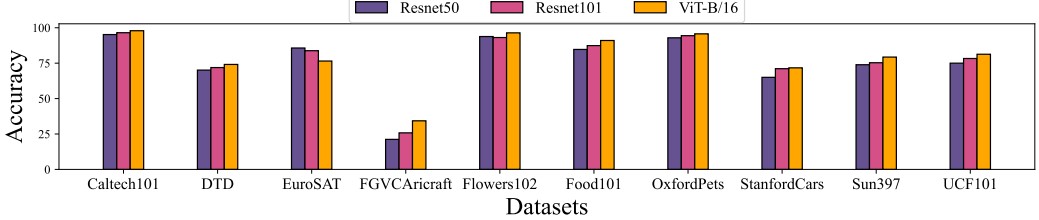

Figure 4: **Ablation of different vision encoder backbones with respect to base accuracy**. A more over-parameterized model leads to better performance across all datasets except EuroSAT.

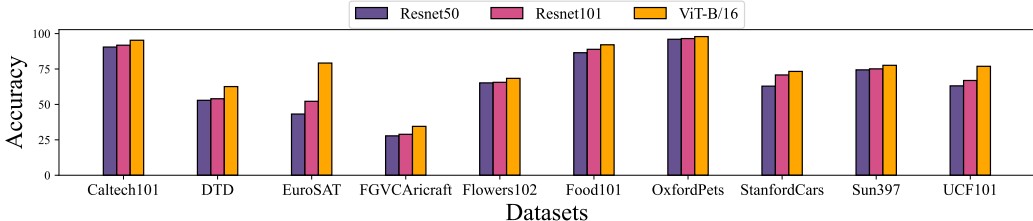

Figure 5: **Ablation of different vision encoder backbones with respect to new accuracy**. A more over-parameterized model leads to better generalization performance across all datasets.

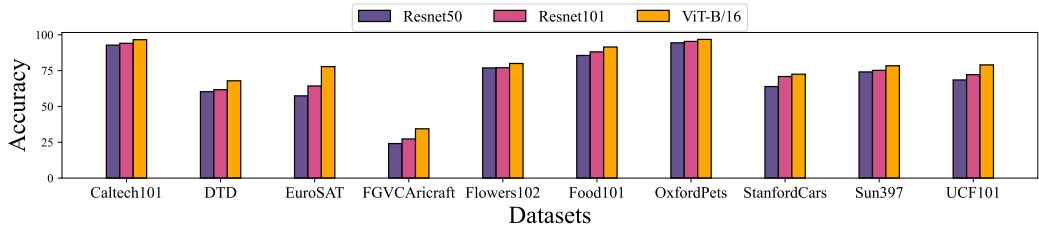

Figure 6: **Ablation of different vision encoder backbones with respect to harmonic mean**. A more over-parameterized model leads to better performance across all datasets.

### A.2.2 BASE AND NEW CLASSES ACCURACY TRADE-OFF

In this section, we ablate whether there is a trade-off between base and new classes accuracy. To do so, we train our proposed method on EuroSAT dataset for 60 and 80 epochs, and compare them with in Table 8. As shown, training EuroSAT for 20 more epochs raises the performance on base classes by $+5.37$. This alone improves the average performance on base classes across the 11 datasets by $+0.49$, resulting in $80.58$ (now +0.11 over CoCoOp). This change slightly affects average harmonic mean, reducing it by $0.17\%$ (now $+1.43\%$ better than CoCoOp). Consequently, we indeed observe a trade-off between performance on base and on new classes. Training for longer will increase the performance on base classes and lower it on new classes.

Table 8: **Base and new classes accuracy trade-off** for EuroSAT dataset. As reported, We indeed observe a trade-off between performance on base and on new classes. Training for longer will increase the performance on base classes and lower it on new classes.

| Dataset | | CoCoOp | CoCoOp+VPT (60) | CoCoOp+VPT* (80) |
|---------|------|--------|------------------|--------------------|
|         | Base | 80.47  | 80.10 ($-0.37$)  | 80.58 ($+0.11$)   |
| Average | New  | 71.69  | 74.94 ($+3.25$)  | 74.22 ($+2.53$)   |
|         | H    | 75.83  | 77.43 ($+1.60$)  | 77.26 ($+1.43$)   |
|         | Base | 87.49  | 80.30 ($-7.19$)  | 85.67 ($-1.82$)   |
| EuroSAT | New  | 60.04  | 75.30 ($+15.26$) | 67.30 ($+7.39$)   |
|         | H    | 71.21  | 77.71 ($+6.50$)  | 75.46 ($+4.25$)   |

### A.2.3 ENSEMBLE BASELINES

Variational prompt tuning can be considered as an efficient ensemble approach as it generates several samples from the prompt space and uses them in conjunction. Nonetheless, we compared CoCoOp+VPT against a modified CoCoOp that naively uses an ensemble of prompts. We implement this by initializing several learnable prompts per class using $M$ hand-crafted templates provided in the CLIP codebase [2] and fine-tuning them together. The final text feature representation is computed as a simple average of the ensemble of prompts for each class as in Section 3.1.4 of (Radford et al., 2021). For a fair comparison, the number of ensembles, $M$, equates to the number of Monte Carlo samples $K$ per dataset (see Table 7). In Table 9, when comparing CoCoOp and CoCoOp + Ens, the ensemble counterpart, we can see that adding an ensemble of prompts does not necessarily lead to an improvement in all datasets. For instance, on DTD and UCF101 datasets, CoCoOp performs better, while on Flowers102, EuroSAT, and FGVCAircraft datasets, CoCoOp + Ens has a higher harmonic mean. Moreover, when comparing CoCoOp + Ens against CoCoOp+VPT , we can see that our method performs better in terms of new classes accuracy and harmonic mean in all datasets other than Flowers102, with an average increase of 1.94% in harmonic mean, whereas the drop in Flower102 is only 0.66%.

### A.2.4 FACTOR OF VARIATION ANALYSIS

Here, we analyze whether prompts that are sampled at the distributional modes correlate with features that characterize any known factors of variation (e.g., sub-domains) in a dataset. We perform this experiment on our best-performing model, CoCoOp+VPT.

First, we randomly select three classes $c_1$, $c_2$, and $c_3$ from the Flowers102 dataset (Columbine, Passion Flower and Cyclamen, for reference). Then, for each class, we compute image features by forwarding the respective input images $\mathbf{x}$ into the image encoder of the CLIP. Next, we apply K-means to group the image features into $Q = 5$ cluster centroids. Note that each cluster centroid

---

[2]https://github.com/openai/CLIP

Table 9: **Comparison between CoCoOp, CoCoOp + Ensembling, and CoCoOp+VPT.** As shown, variational prompt tuning performs better than other alternatives on 4 out of 5 datasets in terms of new accuracy and harmonic mean.

| Dataset | | CoCoOp | CoCoOp+Ens | CoCoOp+VPT |
|---|---|---|---|---|
| DTD | Base | **77.01** | 77.03 | 75.30 |
| | New | 56.00 | 52.17 | **60.80** |
| | H | 64.85 | 62.21 | **67.27** |
| Flowers102 | Base | 94.87 | 94.33 | **95.70** |
| | New | 71.75 | **72.17** | 70.40 |
| | H | 81.71 | **81.78** | 81.12 |
| EuroSAT | Base | **87.49** | 86.60 | 80.30 |
| | New | 60.04 | 66.10 | **75.30** |
| | H | 71.21 | 74.97 | **77.71** |
| FGVCAircraft | Base | 33.41 | **34.93** | 34.40 |
| | New | 23.71 | 32.63 | **35.00** |
| | H | 27.74 | 33.74 | **34.69** |
| UCF101 | Base | 82.33 | 81.20 | **82.53** |
| | New | 73.45 | 73.90 | **75.77** |
| | H | 77.64 | 77.38 | **79.00** |

is assumed to capture some factors of visual variation within each class. Afterwards, we treat the cluster assignments as pseudo-labels of those images. Note that this is done separately for each class. Additionally, for an input image $\mathbf{x}$, we generate $K = 21$ different prompts. These prompts are fed into the clip's text encoder, which generates $K$ different weights $\mathbf{w}_1, \mathbf{w}_2, \cdots, \mathbf{w}_K$ per image instance. We now have a set of $K$ weights for $Q$ pseudo-labels across three classes, which are visualized in each row of Figure 7. In each row, on the left side, we visualize how prompt samples are distributed across the five different clusters, while on the right side, we provide the top-3 most representative samples of the individual clusters (i.e. the images that produce feature representations that are the closets to the centroids). From this figure, it can be seen that there is a high intersection between the distribution of all sub classes, representing shared knowledge. However, some particular visual differences are expressed in regions without intersection in the distribution. Consequently, we believe that there is a correlation between the modes of the variational distribution and visual particularities of subsets of the same class.

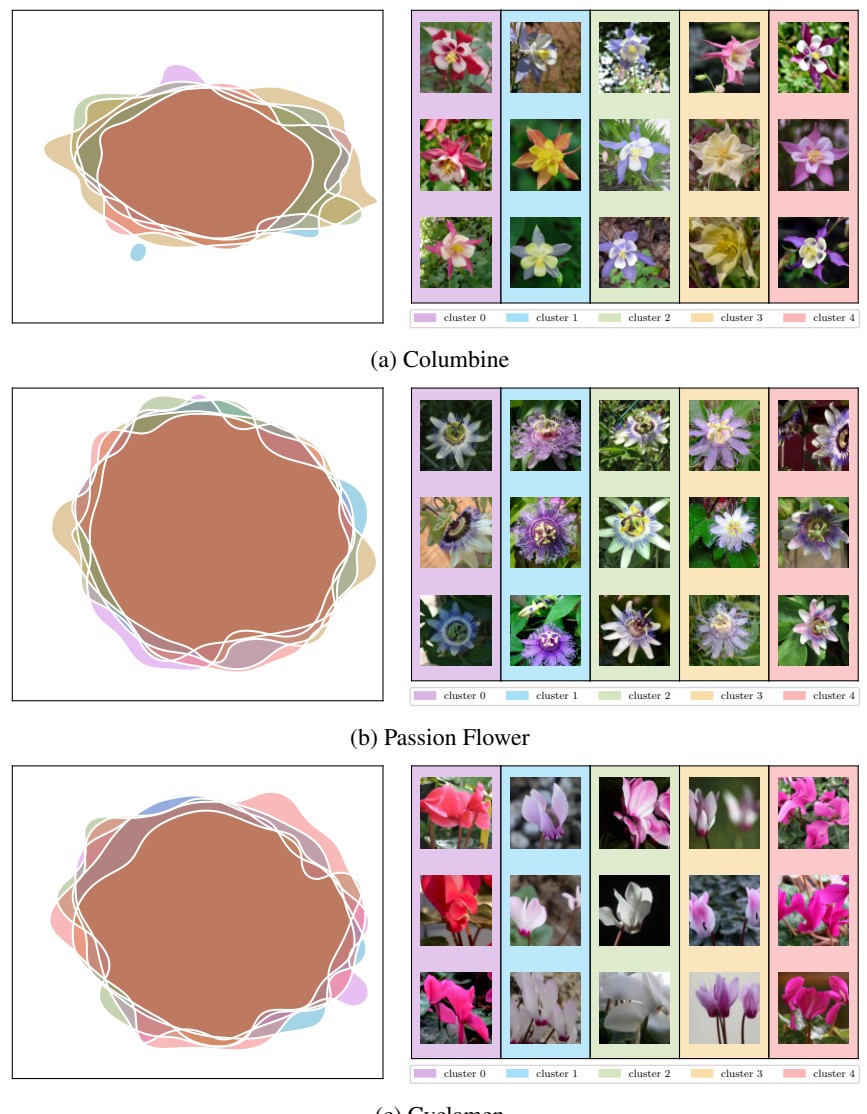

(a) Columbine

(b) Passion Flower

(c) Cyclamen

Figure 7: **Factor of variation analysis**. For three different classes, the contours intersect in a region that represents shared knowledge related to their corresponding class, while diverging a bit and expressing a specific factor of variation. Consequently, we believe that there is a correlation between the modes of the variational distribution and characteristics that adequately represent all known drivers of variation in a dataset (see the text for more details).

### A.2.5 INTERPRETATION OF VARIATIONAL DISTRIBUTION

We now provide an intuition of how prompt samples are distributed within the variational distribution. We perform two experiments on our best-performing model, CoCoOp+VPT.

**Experiment 1:** For an input image $\mathbf{x}$ and its corresponding target $y$, $K$ residuals are sampled from the conditional distribution $\pi_\phi(\mathbf{x})$, which are used to generate $K$ different prompts per class $\mathbf{p}_k = [\mathbf{p}_1 + \mathbf{r}_k, \mathbf{p}_2 + \mathbf{r}_k, \cdots, \mathbf{p}_L + \mathbf{r}_k]$. We also construct a new prompt based on the mean of variational distribution $\mu(\mathbf{x})$ named as mean prompt $\mathbf{p}_{\mu(\mathbf{x})} = [\mathbf{p}_1 + \mu(\mathbf{x}), \mathbf{p}_2 + \mu(\mathbf{x}), \cdots, \mathbf{p}_L + \mu(\mathbf{x})]$. These $K$ prompts are sorted based on their distance to the mean of variational distribution $\mu(\mathbf{x})$, where for any $i$ and $j$ ($i \leq j$), $d(\mu(\mathbf{x}), \mathbf{r}_i) \leq d(\mu(\mathbf{x}), \mathbf{r}_j)$. The $K$ prompts and mean prompt are

prepended to the class-specific embedding to generate a series of $K + 1$ separate classifier weights $\mathbf{w}_{\mu(\mathbf{x}),c}, \mathbf{w}_{1,c}, \mathbf{w}_{2,c}, \cdots, \mathbf{w}_{K,c}$. For each class $y$, we compute the cosine similarity of the image encoding $f(\mathbf{x})$ and $\mathbf{w}_{\mu(\mathbf{x}),y}, \mathbf{w}_{1,y}, \mathbf{w}_{2,y}, \cdots, \mathbf{w}_{K,y}$ and average across all samples per class. In Figure 8, we can see that the mean prompt $\mathbf{p}_{\mu(\mathbf{x})}$ is the most similar classifier weight to the image encoding ■, and, as we move further away from the mean prompt, the cosine similarity decreases ■.

**Experiment 2:** Given weights $\mathbf{w}_{\mu(\mathbf{x}),y}, \mathbf{w}_{1,y}, \mathbf{w}_{2,y}, \cdots, \mathbf{w}_{K,y}$ from Experiment 1, we define a new weight as $\mathbf{w}_y^J = \mathbf{w}_{\mu(\mathbf{x}),y} + \sum_{j=1}^{J} \mathbf{w}_{j,y}$, where $\mathbf{w}_y^J$ is the cumulative sum of weights regarding prompt $\mathbf{p}_1$ to $\mathbf{p}_J$ and mean prompt $\mathbf{p}_{\mu(\mathbf{x})}$. The cosine similarity between the image encoding $f(\mathbf{x})$ and all $\{\mathbf{w}_y^j\}_{j=1}^{j=K}$ are computed and visualized in Figure 8. As shown, summing up classifier's weights together increases the cosine similarity ■, with the maximum similarity obtained when all samples are combined ■.

From these two experiments, we believe that prompt samples are well-distributed inside the prompt space such that the prompt distribution provides adequate coverage of the underlying distribution for downstream tasks.

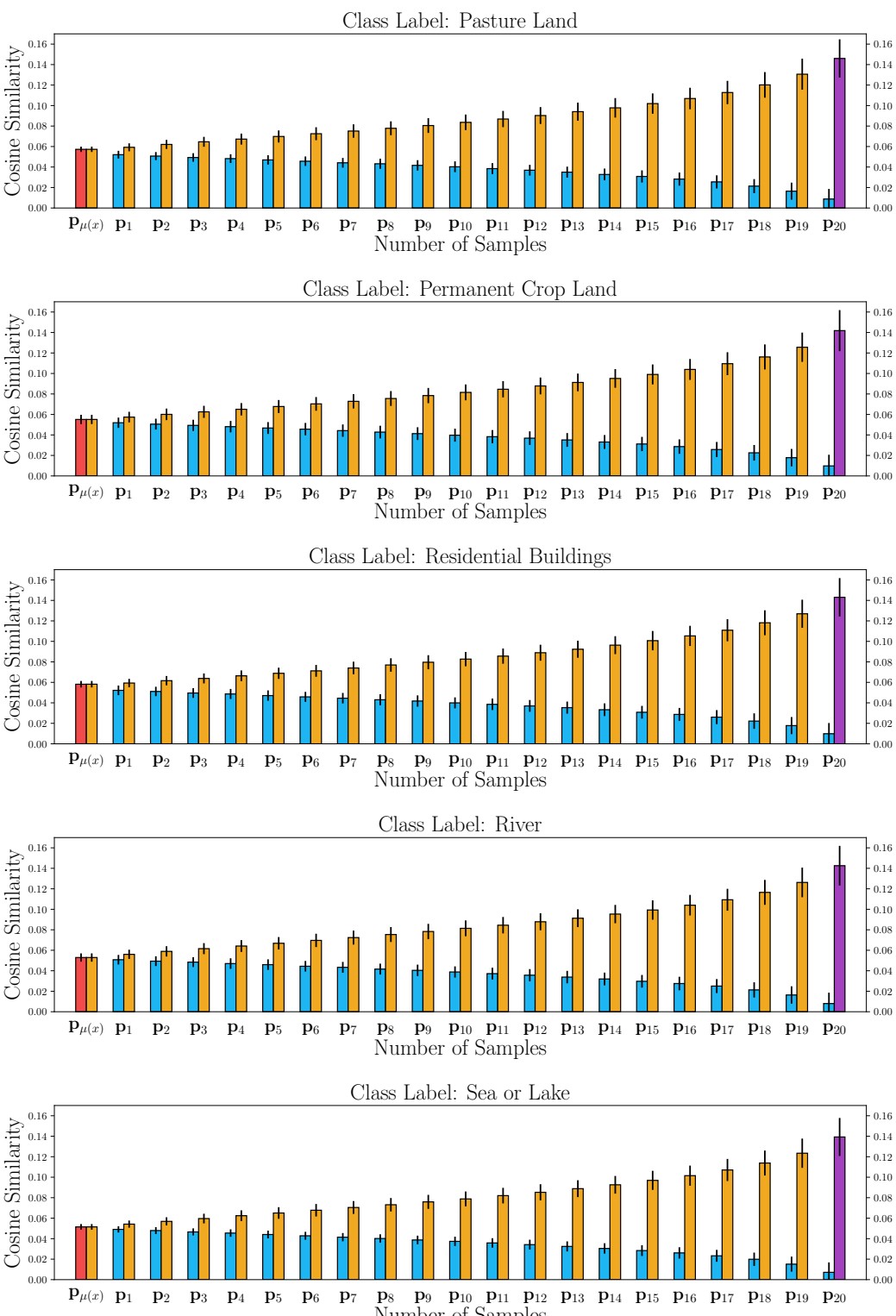

Figure 8: **Interpretation of variational distribution** for EuroSAT dataset. **Experiment 1** ■■ : As shown, the mean prompt $\mathbf{p}_{\mu(\mathbf{x})}$ is the most similar classifier weight to the image embedding, and as we move further away from the mean prompt, the cosine similarity scores decreases. **Experiment 2** ■■ : As shown, suming up classifier's weights increases the cosine similarity, where the maximum similarity is obtained when all classifier weights combined.

