# OpenReview forum: "Variational Prompt Tuning Improves Generalization of Vision-Language Models"
_ICLR.cc/2023/Conference — Submitted to ICLR 2023_

### Official Review · Reviewer_mkLi · 2022-10-25

**Confidence:** 4
**Correctness:** 3
**Technical Novelty And Significance:** 2
**Empirical Novelty And Significance:** 3
**Recommendation:** 6

**Clarity, Quality, Novelty And Reproducibility:**

Clarity: the manuscript is well-written and the majority of claims are well-justified

Quality/Novelty: provides new problem formulation, compared to existing works, albeit somewhat iterative

Reproducibility: no issues

**Strength And Weaknesses:**

(Strengths)

The manuscript proposes a simple, yet principled approach.

The manuscript is well-written and easy to follow.

The manuscript provides numerous experiments, against several strong baselines.

(Weaknesses)

Section 4. The phrase 'Domain Generalization' is slightly misleading, as the section seeks to highlight properties of unseen generalization through zero-shot transfer. The discussion could either be reframed under 'Cross-domain Generalization'.

Section 4. I would like to see further analyses/illustrations of the prompt distribution obtained by VPT. Do prompts that are sampled at the distributional modes correlate with features that well-characterize any known factors of variation (e.g., sub-domains) in the dataset(s)? What further evidence can be provided to show that the prompt distribution approaches reasonable coverage of the underlying distribution? Can equivalent results be obtained through other forms of density estimation?

**Summary Of The Paper:**

The manuscript proposes a probabilistic framework for modeling the underlying distribution of prompts, which acts as an additive module on top of existing prompt-learning approaches, and seeks to enable improved domain generalization.

**Summary Of The Review:**

Missing some intuition regarding the prompt distribution, beyond empirical results

---

> ### Author Response · Authors · 2022-11-18
> **Response to Reviewer mkLi  (1/1)**
>
> Thank you for your review and feedback. We are glad you found our work simple yet principled, well-written, easy to follow, and with numerous experiments against strong baselines. Following your feedback, we added two new subsections A.2.4 and A.2.5 to the appendix of our paper about the ‘Factor of Variation Analysis’ and ‘Interpretation of Variational Distribution’. Below we address your comments.
>
> **The phrase 'Domain Generalization' is slightly misleading, as the section seeks to highlight properties of unseen generalization through zero-shot transfer. The discussion could either be reframed under 'Cross-domain Generalization'.**
>
> The reviewer is right. We have adopted the phrasing as suggested.
>
> **Do prompts that are sampled at the distributional modes correlate with features that well-characterize any known factors of variation (e.g., sub-domains) in the dataset(s)?**
>
> By request of the reviewer, we include the suggested experiment. Indeed, we believe that there is a correlation between the modes of the variational distribution and characteristics that adequately represent drivers of variation in the dataset.
> To do so, we select all samples for a given class label from the Flowers102 dataset and perform K-means clustering on the image features to cluster them according to hidden factors of variation ($Q=5$ cluster centroids). Note that each cluster centroid is assumed to capture some factor of variation within each class. Afterward, we assign each input image to its closest cluster centroid and treat these centroids as pseudo labels for all images. Moreover, for an input image $x$, we generate $K=21$ different prompts. These prompts are fed into the clip's text encoder, which generates $K$ different weights $w_{1}, w_{2}, \cdots, w_{K}$ per image instance. We now have a set of $K$  weights for $Q$ pseudo-labels. We repeat this experiment across three classes, which are visualized in Figure 7.
>
> We added at the end of Section 4.5: “To further aid the interpretation and understanding of the learned prompts, we provide a further ablation on the factor of variation analysis of our best performing model, CoCoOp+VPT in Appendix A.2.4 and A.2.5.”
>
> **What further evidence can be provided to show that the prompt distribution approaches reasonable coverage of the underlying distribution?**
>
> A similar question was raised by Reviewer ANZ7. For completeness, we include here the same reply.
> Since the distribution of the ground truth is unknown, it is hard to say how well variational models match the underlying data distribution. We follow Gordon et al., (2019) and evaluate model fitness based on its average accuracy.
>
> To further explain how the prompt samples are distributed in prompt space, we rely on three different ablations. (1) Effectiveness of the posterior distribution (section 4.5), (2): Number of Monte-Carlo samples (section 4.5), and the newly added  (3) Interpretation of variational distribution; (see A.2.5 in the appendix):
>
> (1) shows that taking one sample from the proposed variational distribution is more useful and informative than taking one sample from a uniform or a normal distribution. We also show that the mean of our proposed variational distribution is the most representative sample.
>
> (2) shows the effect of the number of Monte Carlo samples, where we report that increasing the number of Monte Carlo samples has a positive correlation with the model's performance.
>
> **In the newly designed ablation (3)** (A.2.5 in the appendix), we study the variational distribution of our best-performing model, CoCoOp+VPT.  We do so by plotting in Figure 8 the cosine similarity of the feature representation of the prompts with the image feature representation. We can see that the mean prompt, $p_{mu}(x)$, is the most representative prompt in the space, and that moving away from the mean of the variational distribution reduces the informativeness of the individual prompt samples. Additionally, we also show that the cosine similarity increases as we combine prompts. From these two experiments, we believe that prompt samples are well-distributed inside the prompt space such that the prompt distribution provides adequate coverage of the underlying distribution for downstream tasks.
>
> We added at the end of Section 4.5: “To further aid the interpretation and understanding of the learned prompts we provide a further ablation on the variational distribution of our best performing model, CoCoOp+VPT in Appendix A.2.5.”
>
> **Can equivalent results be obtained through other forms of density estimation?**
>
> We believe that this is indeed possible and have added to the conclusion: “Finally, prompt distribution can be obtained using other ways of density estimation, e.g. normalizing flows, energy-based models, or diffusion models, and we will leave them open for future research.” Thank you.

---

### Official Review · Reviewer_SMRu · 2022-10-25

**Confidence:** 4
**Correctness:** 4
**Technical Novelty And Significance:** 2
**Empirical Novelty And Significance:** 2
**Recommendation:** 6

**Clarity, Quality, Novelty And Reproducibility:**

The paper was well written and was easy to follow overall. More details about zero-shot learning and unconditional setting scenarios would have been useful.

The major ideas used here have already been proposed in existing works, though their adaption and combination is done with simplicity of implementation in mind.

The authors did mention that the code will be released.

**Strength And Weaknesses:**

Pros

- Increase in generalization performance against CoCoOp and ProDA is shown empirically for most datasets, with state-of-the-art results in prompt learning for image recognition tasks with generalization requirements.
- Either of the two mechanisms described - instance specific (conditional prompt learning) and instance independent (standard prompt learning), can be implemented as a simple change to the existing methods with a boost in generalization performance.
- This boost enables higher-performance for few-shot and zero-shot learning for downstream image-text tasks, possibly increasing resilience against distribution shift and adversarial attacks.
- Ablation studies provide evidence of the usefulness of the proposed mechanism.

Cons

- While the proposed model is simple to implement and integrate with existing systems, and has shown improved performance, especially for usage with different domains, there is a concern of lack of enough novelty since the ideas of residual vector and variational prompt learning were already introduced in previous publications.
- The increase in generalization performance comes at a cost to performance of in-domain/seen samples.
- A minor con is that prompt initialization is still manually specified.

Minor Edits

- More details can be specified about implementation in the zero-shot learning scenarios.
- More clear details can be added about the implementation of this method in the unconditional setting of CoOp.
- Can discuss ensembling interpretation of sampling

**Summary Of The Paper:**

This work proposes an instance-specific variational prompt-tuning framework for image-language downstream tasks. Here, the experiments are performed with a frozen CLIP model while fine-tuning the parameters used for prompt creation. It combines two ideas from existing publications on prompt-tuning with learnable prompt embeddings:

1. Adding a residual vector derived from input image features to the learnable prompt embeddings
2. A variational framework for prompt tuning, which this work implements by assuming a Gaussian distribution on residual vectors that are then added to the prompt embeddings.

Implementation of this framework is discussed in both the conditional (instance-specific) setting, as described above, and the unconditional setting of the previous work CoOpOp. Evaluation is performed on 15 image recognition and classification datasets via experiments on the tasks of base-to-new generalization, cross-dataset transfer, and domain generalization.

**Summary Of The Review:**

Borderline accept. The paper presents useful results about the implementation of simple changes to existing systems and the resultant performance improvement in certain scenarios. However, since existing ideas are heavily used and combined, there is an overall lack of novelty, so I would hesitate from a strong accept.

---

> ### Author Response · Authors · 2022-11-18
> **Response to Reviewer SMRu (1/2)**
>
> Thank you for your review and feedback. We appreciate the comments regarding the usefulness and simplicity of our proposed method. Below, we address your other comments.
>
> **While the proposed model is simple to implement and integrate with existing systems, and has shown improved performance, especially for usage with different domains, there is a concern of lack of enough novelty since the ideas of residual vector, and variational prompt learning were already introduced in previous publications.**
>
> A similar question was raised by Reviewer IM4Z. For completeness, we include here the same reply.
>
> To the best of our knowledge, we are the first ones to propose the variational formulation for prompt learning. If the reviewer is referring to ProDa (Lu et al., 2022), it is not a variational formulation. ProDa considers an ensemble of prompts, and then computes the mean and variance of their embedding vectors. They do so without variational formulation and without stochastic sampling. ProDa models the empirical distribution of the output weights after being generated by the learned prompts. Instead, we directly model the distribution of the input prompts.
>
> Regarding the residual addition method (CoCoOp), we do not claim it as a novelty. We just use it as a baseline as it is the current state-of-the-art. Our method does not exploit anything specific to the residual addition, and in the main tables, we show this by combining our method with CoOp (which does not have the residual addition). The performance improvements are very much the same on average.
>
> We will also amend the main text so that our contributions and differences with related work are more clear.
>
> **The increase in generalization performance comes at a cost to the performance of in-domain/seen samples.**
>
> A similar question was raised by Reviewers ANZ7 and IM4Z. For completeness, we include here the same reply.
>
> We indeed observe a trade-off between performance on base and on new classes. Training for longer will increase the performance on base classes and lower it on new classes. This has also been noted in concurrent work, e.g., Table 2 in [a].
>
> Our reported performance on base classes is very similar, although slightly under, to that reported in CoCoOp (-0.37). Picking a slightly longer schedule trivially bridges this performance gap on base classes. For example, training EuroSAT for 20 more epochs raises the performance on base classes by +5.37. This alone improves the average performance on base classes across the 11 datasets by +0.49, resulting in 80.58 (now +0.11 over CoCoOp). This change slightly affects average harmonic mean, reducing it by 0.17% (now +1.43% better than CoCoOp).
>
> | Dataset | CoCoOp | +VPT | +VPT* |
> |----------|----------|-------------|-------------|
> | EuroSAT (base / new / mean) | 87.49 / 60.04 / 71.21 | 80.30 (-7.19) / 75.30 (+15.26) /  77.71 (+6.50) | 85.67 (-1.82) / 67.43 (+7.39) / 75.46 (+4.25)|
> | Average (base / new / mean) | 80.47 / 71.69 / 75.83 | 80.10 (-0.37) / 74.94 (+3.25) / 77.43 (+1.60) | 80.58 (+0.11) / 74.22 (+2.53) / 77.26 (+1.43) |
>
> We have added the Table in the appendix A.2.2 and inserted it into Section 4.2: “Note here however that training for longer increases the performance on base classes and lowers it on new classes. Our setting optimizes harmonic mean, but a simple tweak to the training scheduler on EuroSAT is enough to surpass CoCoOp on base class accuracy, see Table 8 on appendix A.2.2.”
> Thank you.
>
>
> [a] Muhammad Uzair Khattak, Hanoona Rasheed, Muhammad Maaz, Salman Khan, Fahad Shahbaz Khan. MaPle: multi-modal prompt learning. arXiv:2210.03117
>
> **A minor con is that prompt initialization is still manually specified.**
>
> We agree that this is indeed a minor inconvenience. In Table 5, when comparing the two variants of initialization, random against “An image of a {class}”, we indeed show that an appropriately initialized prompt consistently outperforms a randomly initialized prompt. As other methods (e.g., CoOp and CoCoOp) also suffer from the same issue, this might be an interesting future research direction. We updated the paragraph ‘Prompt initialization’ in section 4.5 accordingly.
>
>
> **More details can be specified about implementation in the zero-shot learning scenarios. More clear details can be added about the implementation of this method in the unconditional setting of CoOp.**
>
> We have updated the paper to provide more details as well as cleaner definitions. Moreover, we will release all source code for ease of reproducibility.

---

> > ### Author Response · Authors · 2022-11-18
> > **Response to Reviewer SMRu (2/2)**
> >
> > **Can discuss ensembling interpretation of sampling**
> >
> > A similar question was raised by Reviewer ANZ7. For completeness, we include here the same reply.
> >
> > By request of the reviewer we include the suggested ensemble baseline. Indeed, using an ensemble of prompts on top of CoCoOp does not necessarily lead to improvements. For instance, on DTD and UCF101, CoCoOp performs better, while on Flowers102, EuroSAT, and FGVCAircraft, the ensemble baseline has a higher harmonic mean. Moreover, when comparing the ensemble version of CoCoOp against our proposal, our method performs better in terms of new classes accuracy and harmonic mean on 4 out 5 datasets used in our ablations, with an average increase of 1.94% in harmonic mean. The ensemble baseline results are included in Table 8 and described in appendix A.2.3.
> >
> > Our interpretation: Prompt learning uses few in-domain examples (e.g., 16 examples per class), and thus directly using ensembling will lead to overfitting and poorer generalization. By parameterizing the input space through a Gaussian distribution and stochastic sampling, we reduce overfitting.
> >
> > We added to section 4.2: “We also considered ensemble learning the soft prompt from different ways of initialization, which does not lead to an improvement over our proposal (see Appendix A.2.2).”
> >
> > **In the newly designed ablation (3)** (A.2.5 in the appendix), we study the variational distribution of our best-performing model, CoCoOp+VPT.  We do so by plotting in Figure 8 the cosine similarity of the feature representation of the prompts with the image feature representation. We can see that the mean prompt, $p_{mu}(x)$, is the most representative prompt in the space, and that moving away from the mean of the variational distribution reduces the informativeness of the individual prompt samples. Additionally, we also show that the cosine similarity increases as we combine prompts. From these two experiments, we believe that prompt samples are well-distributed inside the prompt space such that the prompt distribution provides adequate coverage of the underlying distribution for downstream tasks.
> >
> > We added at the end of Section 4.5: “To further aid the interpretation and understanding of the learned prompts we provide a further ablation on the variational distribution of our best performing model, CoCoOp+VPT in Appendix A.2.5.” Thank you.

---

### Official Review · Reviewer_iM4z · 2022-10-25

**Confidence:** 2
**Correctness:** 4
**Technical Novelty And Significance:** 2
**Empirical Novelty And Significance:** 3
**Recommendation:** 6

**Clarity, Quality, Novelty And Reproducibility:**

The paper is clear, and the experiments are easy to follow. The paper has limited technical novelty because it combines two existing ideas of residual connection and variational formulation of prompt learning

**Strength And Weaknesses:**


Strengths:
- The method is intuitive and show strong empirical results on a bunch of tasks. The ablations are reasonable too!
The model maintains generalization capability for both new  and old classes.

Weaknesses:
- The paper combines two existing ideas in prompt learning (the residual addition, and the variational formulation). This limits the technical novelty of the paper a little bit.
- The whole motivation of the paper is about preserving the generalization capability for downstream tasks, yet we see that the model perfoms worse on base classes than existing methods.


**Summary Of The Paper:**

Summary:
The authors propose a variational prompt generator conditioned on an input instance. This is learned by adding a residual (which is input conditioned) to a fixed set of learnt prompts. During inference, multiple residuals can be sampled to generate different prompts. The authors show that the proposed approach surpasses existing methods on 16/20 datasets for classification


**Summary Of The Review:**

I think the paper has merits, but technical novelty is limited. I am on the fence about the paper because it has strong empirical results but the approach combines existing ideas in unsurprising ways. I will look at the author response and read other reviews to decide my final rating.

Update after rebuttal: I think the authors took effort in addressing all my concern and most of other revierwer's concerns during the rebuttal. I am still on the fence due to limited novelty and tradeoff in performance between base and new classes.  But I think the new analysis makes the paper stronger. To reflect this I am updating my score to 6 (marginally above acceptance threshold).

---

> ### Author Response · Authors · 2022-11-18
> **Response to Reviewer iM4z  (1/1)**
>
> Thank you for your review and feedback. We appreciate the comments stating that the method is intuitive, with strong empirical results, and maintains generalization power for downstream tasks. Below, we address your comments.
>
> **The paper combines two existing ideas in prompt learning (the residual addition and the variational formulation). This limits the technical novelty of the paper a little bit.**
>
> To the best of our knowledge, we are the first ones to propose the variational formulation for prompt learning. If the reviewer is referring to ProDa (Lu et al., 2022), it is not a variational formulation. ProDa considers an ensemble of prompts, and then computes the mean and variance of their embedding vectors. They do so without variational formulation and without stochastic sampling. ProDa models the empirical distribution of the output weights after being generated by the learned prompts. Instead, we directly model the distribution of the input prompts.
>
> Regarding the residual addition method (CoCoOp), we do not claim it as a novelty. We just use it as a baseline as it is the current state-of-the-art. Our method does not exploit anything specific to the residual addition, and in the main tables, we show this by combining our method with CoOp (which does not have the residual addition). The performance improvements are very much the same on average.
>
> We will also amend the main text so that our contributions and differences with related work are more clear.
>
> **The whole motivation of the paper is about preserving the generalization capability for downstream tasks, yet we see that the model performs worse on base classes than existing methods.**
>
> A similar question was raised by Reviewer ANZ7. For completeness, we include here the same reply.
>
> We indeed observe a trade-off between performance on base and on new classes. Training for longer will increase the performance on base classes and lower it on new classes. This has also been noted in concurrent work, e.g., Table 2 in [a].
>
> Our reported performance on base classes is very similar, although slightly under, to that reported in CoCoOp (-0.37). Picking a slightly longer schedule trivially bridges this performance gap on base classes. For example, training EuroSAT for 20 more epochs raises the performance on base classes by +5.37. This alone improves the average performance on base classes across the 11 datasets by +0.49, resulting in 80.58 (now +0.11 over CoCoOp). This change slightly affects average harmonic mean, reducing it by 0.17% (now +1.43% better than CoCoOp).
>
> | Dataset | CoCoOp | +VPT | +VPT* |
> |----------|----------|-------------|-------------|
> | EuroSAT (base / new / mean) | 87.49 / 60.04 / 71.21 | 80.30 (-7.19) / 75.30 (+15.26) /  77.71 (+6.50) | 85.67 (-1.82) / 67.43 (+7.39) / 75.46 (+4.25)|
> | Average (base / new / mean) | 80.47 / 71.69 / 75.83 | 80.10 (-0.37) / 74.94 (+3.25) / 77.43 (+1.60) | 80.58 (+0.11) / 74.22 (+2.53) / 77.26 (+1.43) |
>
> We have added the Table in the appendix A.2.2 and inserted it into Section 4.2: “Note here however that training for longer increases the performance on base classes and lowers it on new classes. Our setting optimizes harmonic mean, but a simple tweak to the training scheduler on EuroSAT is enough to surpass CoCoOp on base class accuracy, see Table 8 on appendix A.2.2.”
> Thank you.
>
>
> [a] Muhammad Uzair Khattak, Hanoona Rasheed, Muhammad Maaz, Salman Khan, Fahad Shahbaz Khan. MaPle: multi-modal prompt learning. arXiv:2210.03117

---

### Official Review · Reviewer_ANZ7 · 2022-10-31

**Confidence:** 4
**Correctness:** 3
**Technical Novelty And Significance:** 2
**Empirical Novelty And Significance:** Not applicable
**Recommendation:** 5

**Clarity, Quality, Novelty And Reproducibility:**

Clarity, quality and reproducibility look overall good. The novelty is rather limited.

**Strength And Weaknesses:**

Pros:
1. The proposed idea is intuitive and straightforward.
2. The improvement is shown by combing the proposed method with two popular methods for visual prompt tuning.

Cons:
1. It is unclear why it is needed to learn the distribution of prompts. Unlike where variational inference was usually used in computer vision, e.g., VAE, the possible distribution of the prompts is actually not that untractable.
a. Simple baselines like ensemble learning the soft prompt from different ways of initialization should be provided. It is possible that when ensembling several prompts learned from different initializations, the performance of the model is already somewhat achieved. Because the possible way of introducing an image in caption is rather limited. Probably a few main ways of the prompt initialization can already cover.
b. Any interpretation of the learned/sampled prompts could be provided? This may help to understand why the learning process is necessary.
c. Although the authors claim on the improvement of generalization to new classes, the performance gain on cross-dataset setting is still quite limited. Therefore, the model may still learns pretty domain specific distribution of prompts.

2. In Table1, more than half pf the results sacrifice the base class accuracy for improvement of the average. Therefore, it is also important to provide simple baselines like early stopping or adding some common regularization terms on the prompts.


**Summary Of The Paper:**

In this paper, the authors proposed to leverage variational inference to learn the underlying distribution of the prompts with or without image feature as the condition. The effectiveness of the proposed method is verified on several benchmarks.

**Summary Of The Review:**

Overall, the reviewer thinks this paper solves an incremental problem following a rather standard approach without providing in-depth analysis and insights on why solving this problem.

---

> ### Author Response · Authors · 2022-11-18
> **Response to Reviewer ANZ7 (1/2)**
>
> Thank you for your review and feedback. We appreciate the comments about the proposed idea being intuitive and straightforward and the recognition that it offers consistent improvements.
>
> Following the reviewer’s feedback, we have now added two new subsections to the appendix, A.2.3 and A.2.5. These include new experiments on the comparison with **a baseline ensemble version of CoCoOp** and **Interpretation of Variational Distribution**. Below, we include these experiments for completeness, and we address the individual comments.
>
> **It is unclear why it is needed to learn the distribution of prompts. Unlike where variational inference was usually used in computer vision, e.g., VAE, the possible distribution of the prompts is actually not that intractable.**
> **a. Simple baselines like ensemble learning the soft prompt from different ways of initialization should be provided. It is possible that when ensembling several prompts learned from different initializations, the performance of the model is already somewhat achieved. Because the possible way of introducing an image in a caption is somewhat limited. Probably a few main ways of prompt initialization can already cover.**
>
> By request of the reviewer we include the suggested ensemble baseline. Indeed, using an ensemble of prompts on top of CoCoOp does not necessarily lead to improvements. For instance, on DTD and UCF101, CoCoOp performs better, while on Flowers102, EuroSAT, and FGVCAircraft, the ensemble baseline has a higher harmonic mean. Moreover, when comparing the ensemble version of CoCoOp against our proposal, our method performs better in terms of new classes accuracy and harmonic mean on 4 out 5 datasets used in our ablations, with an average increase of 1.94% in harmonic mean. The ensemble baseline results are included in Table 8 and described in appendix A.2.3.
>
> Our interpretation: Prompt learning uses few in-domain examples (e.g., 16 examples per class), and thus directly using ensembling will lead to overfitting and poorer generalization. By parameterizing the input space through a Gaussian distribution and stochastic sampling, we reduce overfitting.
>
> We added to section 4.2: “We also considered ensemble learning the soft prompt from different ways of initialization, which does not lead to an improvement over our proposal (see Appendix A.2.3).”
>
> **Any interpretation of the learned/sampled prompts could be provided? This may help to understand why the learning process is necessary.**
>
> Since the distribution of the ground truth is unknown, it is hard to say how well variational models match the underlying data distribution. We follow Gordon et al., (2019) and evaluate model fitness based on its average accuracy.
>
> To further explain how the prompt samples are distributed in prompt space, we rely on three different ablations. **(1)** Effectiveness of the posterior distribution (section 4.5), **(2)**: Number of Monte-Carlo samples (section 4.5), and the newly added  **(3) Interpretation of variational distribution**; (see A.2.5 in the appendix):
>
> **(1)** shows that taking one sample from the proposed variational distribution is more useful and informative than taking one sample from a uniform or a normal distribution. We also show that the mean of our proposed variational distribution is the most representative sample.
>
> **(2)** shows the effect of the number of Monte Carlo samples, where we report that increasing the number of Monte Carlo samples has a positive correlation with the model's performance.
>
> **In the newly designed ablation (3)** (A.2.5 in the appendix), we study the variational distribution of our best-performing model, CoCoOp+VPT.  We do so by plotting in Figure 8 the cosine similarity of the feature representation of the prompts with the image feature representation. We can see that the mean prompt, $p_{mu}(x)$, is the most representative prompt in the space, and that moving away from the mean of the variational distribution reduces the informativeness of the individual prompt samples. Additionally, we also show that the cosine similarity increases as we combine prompts. From these two experiments, we believe that prompt samples are well-distributed inside the prompt space such that the prompt distribution provides adequate coverage of the underlying distribution for downstream tasks.
>
> We added at the end of Section 4.5: “To further aid the interpretation and understanding of the learned prompts we provide a further ablation on the variational distribution of our best performing model, CoCoOp+VPT in Appendix A.2.5.”

---

> > ### Author Response · Authors · 2022-11-18
> > **Response to Reviewer ANZ7 (2/2)**
> >
> > **Although the authors claim on the improvement of generalization to new classes, the performance gain on cross-dataset setting is still quite limited. Therefore, the model may still learns pretty domain specific distribution of prompts.**
> >
> > Indeed, we would expect that prompt distribution generalizes better within domain than across domains. The base-to-new task refers to generalization within a single domain, whereas the cross-dataset transfer learning is across two domains, where the underlying distributions are less likely to transfer. However, on target datasets our method improves on average 1.63% over CoOp and 0.16% over CoCoOp. From another perspective, our proposal improves over CoOp in 9 out of 10 datasets and over CoCoOp for 7 out of 10 datasets.
> >
> > **In Table1, more than half of the results sacrifice the base class accuracy for improvement of the average. Therefore, it is also important to provide simple baselines like early stopping or adding some common regularization terms on the prompts.**
> >
> > We indeed observe a trade-off between performance on base and on new classes. Training for longer will increase the performance on base classes and lower it on new classes. This has also been noted in concurrent work, e.g., Table 2 in [a].
> >
> > Our reported performance on base classes is very similar, although slightly under, to that reported in CoCoOp (-0.37). Picking a slightly longer schedule trivially bridges this performance gap on base classes. For example, training EuroSAT for 20 more epochs raises the performance on base classes by +5.37. This alone improves the average performance on base classes across the 11 datasets by +0.49, resulting in 80.58 (now +0.11 over CoCoOp). This change slightly affects average harmonic mean, reducing it by 0.17% (now +1.43% better than CoCoOp).
> >
> > | Dataset | CoCoOp | +VPT | +VPT* |
> > |----------|----------|-------------|-------------|
> > | EuroSAT (base / new / mean) | 87.49 / 60.04 / 71.21 | 80.30 (-7.19) / 75.30 (+15.26) /  77.71 (+6.50) | 85.67 (-1.82) / 67.43 (+7.39) / 75.46 (+4.25)|
> > | Average (base / new / mean) | 80.47 / 71.69 / 75.83 | 80.10 (-0.37) / 74.94 (+3.25) / 77.43 (+1.60) | 80.58 (+0.11) / 74.22 (+2.53) / 77.26 (+1.43) |
> >
> > We have added the Table in the appendix A.2.2 and inserted it into Section 4.2: “Note here however that training for longer increases the performance on base classes and lowers it on new classes. Our setting optimizes harmonic mean, but a simple tweak to the training scheduler on EuroSAT is enough to surpass CoCoOp on base class accuracy, see Table 8 on appendix A.2.2.”
> > Thank you.
> >
> >
> > [a] Muhammad Uzair Khattak, Hanoona Rasheed, Muhammad Maaz, Salman Khan, Fahad Shahbaz Khan. MaPle: multi-modal prompt learning. arXiv:2210.03117

---

### Author Response · Authors · 2022-12-05
**A friendly reminder of the rebuttal conclusion**

Thank you again for your thoughtful comments and constructive suggestions. We have provided thorough responses and additional experimental results. We would appreciate it if you could read our responses and inform us and the other reviewers if your concerns have been addressed. We are glad to further discuss any remaining concerns that you find not fully addressed. Thank you.

Best regards, Authors

---

### Decision · Program_Chairs · 2023-01-20

**Decision:**

Reject

**Justification For Why Not Higher Score:**

The paper lacks an explanation of why the in-domain (base) performance is significantly dropped after applying the proposed method. Therefore, the paper is not yet ready for publication.

**Justification For Why Not Lower Score:**

N/A

**Metareview: Summary, Strengths And Weaknesses:**

This paper presents a variational learning method for prompt tuning for few-shot image classification. Like any variational Bayes method, it assumes that the visual prior can be encoded into a Gaussian prior and so the objective is to minimize the ELBO. The method can be plugged into existing prompt tuning methods like CoOp and CoCoOp.

Strength:
Clear writing and comprehensive experiments.

Weakness:
The paper doesn't provide insights into why such a variational layer can improve the overall generalization. In fact, a critical concern, as raised by all the reviewers, the significant loss of in-domain (base) performance is somewhat unacceptable, which may imply severe theoretical flaws of the proposed method.

**Summary Of Ac-Reviewer Meeting:**

Three reviewers are marginally positive and one is marginally negative. They share the same concern that the in-domain (base) performance is significantly dropped by using the proposed method. After rebuttal, the authors failed to explain the reasons. AC read the paper, reviews, and rebuttal, and did find that this weakness is very critical, which may undermine the claims of the paper. Therefore, the paper is not yet ready for publication.